# Dose-Dependent Effect of Mitochondrial Superoxide Dismutase Gene Overexpression on Radioresistance of HEK293T Cells

**DOI:** 10.3390/ijms242417315

**Published:** 2023-12-10

**Authors:** Marina M. Tavleeva, Elena E. Rasova, Anna V. Rybak, Elena S. Belykh, Elizaveta A. Fefilova, Elizaveta M. Pnachina, Ilya O. Velegzhaninov

**Affiliations:** 1Institute of Biology of Komi Scientific Centre, Ural Branch of Russian Academy of Sciences, 28b Kommunisticheskaya St., Syktyvkar 167982, Russia; tavleeva.marina@yandex.ru (M.M.T.); rasova@ib.komisc.ru (E.E.R.); canewa.anuta@yandex.ru (A.V.R.); daryd22@gmail.com (E.S.B.); 2Institute of Cytology, Russian Academy of Sciences, 4 Tikhoretsky Ave., St. Petersburg 194064, Russia; st093552@student.spbu.ru; 3Institute of Biology and Biomedicine, Lobachevsky State University of Nizhny Novgorod, 23 Gagarin Ave., Nizhny Novgorod 603950, Russia; sova.zever@gmail.com

**Keywords:** *SOD2*, *CAT*, overexpression dose, CRISPRa, cellular radioresistance, transcript variants

## Abstract

Over the last two decades, a multitude of gain-of-function studies have been conducted on genes that encode antioxidative enzymes, including one of the key enzymes, manganese superoxide dismutase (*SOD2*). The results of such studies are often contradictory, as they strongly depend on many factors, such as the gene overexpression level. In this study, the effect of altering the ectopic expression level of major transcript variants of the *SOD2* gene on the radioresistance of HEK293T cells was investigated using CRISPRa technology. A significant increase in cell viability in comparison with the transfection control was detected in cells with moderate *SOD2* overexpression after irradiation at 2 Gy, but not at 3 or 5 Gy. A further increase in the level of *SOD2* ectopic expression up to 22.5-fold resulted in increased cell viability detectable only after irradiation at 5 Gy. Furthermore, a 15–20-fold increase in *SOD2* expression raised the clonogenic survival of cells after irradiation at 5 Gy. Simultaneous overexpression of genes encoding SOD2 and Catalase (CAT) enhanced clonogenic cell survival after irradiation more effectively than separate overexpression of both. In conjunction with the literature data on the suppression of the procarcinogenic effects of superoxide dismutase overexpression by ectopic expression of *CAT*, the data presented here suggest the potential efficacy of simultaneous overexpression of *SOD2* and *CAT* to reduce oxidative stress occurring in various pathological processes. Moreover, these results illustrate the importance of selecting the degree of *SOD2* overexpression to obtain a protective effect.

## 1. Introduction

Redox balance disruption caused by an excess of reactive oxygen species (ROS) and/or inefficient antioxidant systems leads to oxidative stress [1]. On the other hand, superoxide anions (O_2_^•−^), products of the mitochondrial electron transport chain, as well as products of reactions involving it, are signaling molecules [2]. As a consequence, disbalance in O_2_^•−^ plays important roles in the pathogenesis of a number of diseases [3,4]. Oxidative stress more broadly causes and/or accompanies many pathological conditions and diseases, including cancer [5,6] and neurodegenerative [6,7,8] and cardiovascular diseases [9]. In this regard, it seems obvious that there is a need to accumulate knowledge about the capacity and consequences of managing cellular antioxidant systems [10]. It would be fair to assume that a reasonable increase in the activity of antioxidant systems could be an effective tool for the therapy of diseases that develop or are complicated by oxidative stress. Targeted regulation of stress response gene transcription in human cells can currently be considered as a powerful tool for managing cellular resistance to oxidative stress [11,12]. The development of transcriptional programming technologies, such as CRISPR activation (CRISPRa) and CRISPR interference (CRISPRi), provides great opportunities to uncover mechanisms of human diseases and to develop new therapeutic strategies for their prevention and treatment [11,12,13].

Mitochondrial superoxide dismutase (Mn-SOD, SOD2), a key enzyme for O_2_^•−^ detoxification, plays an important role in the development of resistance to oxidative stress in normal/healthy tissues [4,14,15]. A lack of *SOD2* causes significant impairment of cellular and organismal function. In particular, knockdown or knockout of *SOD2* resulted in mitochondria damage and dysfunction in in vitro [16,17] and in vivo [18,19,20] studies. Furthermore, *SOD2* depletion led to a decrease in the proliferative activity of murine myoblasts [21], to metabolic dysfunction (succinate accumulation and inhibition of the activity of the ten-eleven translocase family of DNA demethylase enzymes) and to aberrant nuclear DNA methylation in human erythroleukemia cells [22]. Knockout of *SOD2* in mice resulted in embryonic or early neonatal lethality [23]. In contrast, overexpression of *SOD2* in many experimental systems leads to an increase in stress tolerance at the cellular and organismal levels. For instance, a decrease in radiation-induced oxidative stress as well as increased expression of genes involved in the radiation adaptive response has been shown in various cell lines [14,24,25,26]. Additionally, *SOD2* overexpression enhanced cellular resistance to gamma radiation [27,28] and to the pro-oxidant antitumor drug 2-Methoxyestradiol [29]. In experiments in vivo, *SOD2* overexpression in certain tissues decreased ionizing irradiation-induced lung inflammation [30] and improved overall cardiac function [31]. Injection of bone marrow-derived mesenchymal stem cells with *SOD2* overexpression resulted in more efficient liver repair in mice after ischemia/reperfusion injury [32] than injection of the same cells without overexpression.

However, in some experimental systems and by some parameters *SOD2* overexpression did not result in an increase in stress resistance. For example, radioresistance was not increased in cervical carcinoma cells (HeLa) [33]. Increasing the expression level and activity of the SOD2 protein at the organismal level failed to alter the resistance of *Drosophila melanogaster* to 100% O_2_ as well as to starvation conditions and resulted in a decrease in lifespan [34].

The effect of *SOD2* on cancer initiation and progression is much more controversial. Increased expression of this gene led to tumor promotion in some experimental systems [35,36,37,38,39,40,41,42,43,44,45,46,47,48] and tumor suppression [47,49,50,51,52] or even prevention of malignancy in normal cells or tissues in others [53,54,55,56,57,58,59]. Meanwhile, the protumorogenic effect of *SOD2* overexpression is associated with H_2_O_2_ overproduction and in some cases can be suppressed by simultaneous *CAT* overexpression [41,42,43,44,45,46].

The diverse and, in some cases, contradictory results of gain-of-function studies of one and the same gene can be explained by a number of reasons, among which are (a) differences in the intensity and direction of effects at various ectopic expression levels, which are hardly taken into account in the majority of such studies; (b) functional differences in protein isoforms encoded by different gene transcript variants; (c) the epigenetic and functional context in which overexpression occurs; (d) signaling and effector multifunctionality of most stress response proteins; (e) the fact that in the case of tumor-related endpoints, the stage of tumor development plays an important role. These reasons are discussed in more detail in the systematic review by Tavleeva et al., 2022 [10].

Comprehension of the factors listed above in terms of their separate action can help in the development of approaches to rational control of the activity of cellular antioxidant systems within the framework of gene therapy and genetic engineering in general. The necessity of studying the dependence of the observed effect on the level of gene overexpression within the framework of any gain-of-function studies seems to be the most obvious, since such dependence is hardly predictable, and the effects may even be multidirectional [60].

Based on the above, the present study aimed to investigate the dependence of human embryonic kidney cells’ (HEK293T) radioresistance on the level of *SOD2* gene overexpression. To achieve this aim, a feature of some variants of CRISPRa technology was employed which permits simple regulation of the gene of interest’s overexpression level by varying the distance between the single guide RNA (sgRNA) recognition site and transcription start site (TSS) as well as the number of concurrently utilized sgRNAs [61]. In the case of *SOD2*, the study of the effects to be observed at different levels of gene expression is complicated by the diversity of transcriptional variants of this enzyme. This led us to the need for a preliminary assessment of the basal and radiation-induced expression levels of the main transcript variants.

## 2. Results and Discussion

### 2.1. Basal Expression Level of SOD2 Transcript Variants

Like many genes in the human genome, *SOD2* has a complex transcriptional map. The splice variants 1–6, which encode four protein isoforms (A, B—3,4, C—5 and D—6), share a TSS; variants 8 and 9, which encode isoform E, have two TSSs that are located 434 nucleotide pairs apart from each other and more than 33,000 nucleotide pairs away from the first one; variant 7, which also encodes isoform E, has another TSS that is more than 69,000 nucleotide pairs away from the first one [62]. This suggests that the expression of splice variants encoding isoform E may be regulated differently from the expression of splice variants encoding isoforms A, B, C and D. The promoter functional annotation and regulation of expression has been studied in depth for the transcription variant that encodes isoform A, which is considered to be the major one [4,63]. The transcriptional map suggests that the regulation of isoforms B–D is similar to that of isoform A. However, the transcriptional regulation, function and activity of isoform E have not yet been studied. Therefore, the first step was to determine the contribution of isoform E and the corresponding transcription variants 7–9 to the basal level of *SOD2* expression. In the case of transcription variants 1–5, which encode isoforms A, B and C, it is not possible to select individual primers; however, the primers specific to transcript groups were designed. The list of the primers used and their schematic arrangement are shown in Figure 1 and Table 1.

The RT-PCR results for HEK293T, HeLa, and human colorectal adenocarcinoma HT-29 cell lines (Figure 2) indicates the prevalence of isoforms that have a common TSS with isoform A (A–C). Isoform D is the only exception. Its mRNA level displays a significant reduction. The transcription activity of TSS for variants 8 and 9, which encode protein isoform E, is roughly three orders of magnitude lower in comparison with variants 1–5 in the HEK293T and HeLa cell lines and two orders of magnitude lower in HT-29 cells (refer to Figure 2B). It appears that transcription variant 7, which has an alternative TSS, is not expressed in these cell lines. This is supported by PCR using four different primer variants, each with a left primer specific to the first exon, which did not yield an amplicon of the expected size (as shown in Figure 2A and Table 1).

### 2.2. Radiation-Induced Expression of SOD2 Transcript Variants

Clearly, both basal and oxidative stress-induced levels of gene transcript expression can provide insight into their functional significance. It is widely acknowledged that ionizing radiation triggers cellular oxidative stress [64], even at low doses [65]. The expression of *SOD2* transcript variants was analyzed in three cell lines at 2–48 h after irradiation at 2 Gy.

As anticipated, a tendency to increase *SOD2* expression was observed in the cells analyzed using primers that amplify all recognized transcripts of the gene (alltv) after irradiation. However, it is important to note that a significant increase in expression level was only observed in HeLa cells (Figure 3). Nevertheless, the contribution of different transcriptional variants was ambiguous. The expression of transcripts 1–4 (isoforms A and B) as well as 1, 2 and 5 (isoforms A and C) rose in HEK293T and HeLa cells after 24 and 48 h, respectively. Conversely, an observable decrease in *SOD2* expression level occurred in HT29 cells after 24 h. The role of transcript variant 5 in inducing *SOD2* expression was only detected in the HeLa and HEK293T cell lines. In HT29 cells, transcript variant 8, which encodes the E isoform of the SOD2 protein, exhibited the highest induction of expression in response to radiation. An increase in the expression level of transcript variants 8 and 9, induced by radiation, was observed in both HeLa and HEK293T cells. The expression of transcript variant 6, which encodes isoform D, displayed an increasing tendency 48 h post-irradiation of HT29 and HEK293T cells, without reaching a statistically significant change.

Considering the high basal level of expression of transcript variants 1–5, and their inducibility by irradiation, we chose them to evaluate the effects of varied levels of ectopic *SOD2* expression on cellular radioresistance. At the same time, the identification of radiation induction of transcript variants 8 and 9 suggests that the significance of isoform E may have been underestimated. Further investigation is necessary since mitochondrial superoxide dismutase is crucial for the cellular antioxidant defense.

### 2.3. Effects of Different Degrees of SOD2 Overexpression on Cellular Radioresistance

To achieve the overexpression of transcript variants 1–6 of the *SOD2* gene through CRISPRa, guide RNAs were designed using 300-nucleotide sequences upstream of the corresponding TSS. The location of the guide RNAs in the gene transcription map and their sequences are shown in Figure 1 and Table 2, respectively. Using variants of simultaneous co-transfection with all three plasmids encoding sgRNAs, only the two closest to the TSS (sgRNA1 and sgRNA2), or only the proximal (sgRNA1) and distal (sgRNA3) plasmids separately, varying degrees of overexpression of the *SOD2* gene were achieved (1.5- to 22.5-fold) (Figure 4A). Expression changes were observed only in transcript variants 1–6, while the expression of transcript variants 8 and 9, which have a transcriptional start site significantly upstream of the sgRNA landing sites, remained unchanged.

Cells that were co-transfected with the activator plasmid and plasmids encoding only sgRNA1 demonstrated significantly higher resistance to 2 Gy irradiation compared to cells subjected to control transfection (Figure 4B). In this transfection variant, the overexpression of *SOD2* was relatively weak (3.7–4.6-fold) (Figure 4A). Following irradiation of transfected cells at a dose of 3 Gy, there were no statistically significant differences in cell viability observed in the fluorometric microculture cytotoxicity assay (FMCA). However, there was a tendency for increased cell radioresistance in the variant which underwent simultaneous transfection with plasmids encoding sgRNA1 and sgRNA2 (11.1–14.0-fold *SOD2* overexpression). After irradiation at 5 Gy, all cell cultures exhibiting *SOD2* overexpression in a wide range (3.7- to 22.5-fold) were more viable than non-overexpressing cells. At the same time, the highest level of overexpression, which was attained through the co-transfection of all three sgRNAs, demonstrated a positive impact on cellular viability only at a radiation dose of 5 Gy. However, minimal overexpression (1.4–1.6-fold), achieved by transfecting dCas9-VPH with only sgRNA3, did not affect cell viability under any irradiation conditions.

The obtained result clearly demonstrates that the response to γ-irradiation exposure is different at the various degrees of *SOD2* overexpression. This fact may account for the often inconsistent results of gain-of-function studies, as was demonstrated in human breast cancer MCF-7 cells [60]. A moderate increase in *SOD2* activity (2- to 6-fold compared to the parental line) resulted in the inhibition of Hypoxia-inducible factor-1 (HIF-1α) accumulation in response to hypoxia, and, in contrast, when *SOD2* activity exceeded a 6-fold increase, a rise in HIF-1α accumulation occurred. The authors attribute this phenomenon to the excessive accumulation of H_2_O_2_, which inhibits HIF-1α degradation. Taken together, the presented evidence suggests that the extent of upregulation relative to the baseline level is one of the crucial considerations for effective gene therapy targeting the regulation of oxidative stress.

The variances in cell resistance to ionizing radiation, which result from different levels of *SOD2* overexpression, seemingly arise from the integration of multiple, multidirectional factors that perturb the system’s elements. Hydrogen peroxide, the product of dismutation reactions, is a highly reactive molecule with dual functionality: on the one hand, it induces oxidative stress, while on the other, it serves as a signal that promotes proliferation [66,67,68]. Furthermore, O_2_^•−^, which is the substrate of *SOD2*, is a signaling molecule [3], and the SOD2 protein itself partakes in numerous signaling cascades [15,24,69]. The variety of functions of the protein, its substrate, and the reaction product elucidate the multidirectional effects of changes in the expression of this enzyme in other studies. For instance, Lee et colleagues [21] demonstrated a reduction in the myocyte proliferation rate in transgenic mice with both decreased and increased levels of *SOD2* expression, while the decrease was more significant in animals with overexpression of the gene. Moreover, in most experimental systems, the increased expression of *SOD2* resulted in decreased cell proliferation [70,71], while in some cases, it led to an opposite effect [38,72].

Therefore, to separate the contribution to the obtained dose-dependent effects of cell resistance itself from that of changes in the proliferation rate, an additional experiment was performed using the colony formation assay (CFA). The level of proliferation was assessed by measuring the area of colonies formed by surviving cells. In addition, moderate and maximal overexpression of *SOD2* was performed separately and simultaneously with overexpression of *CAT*.

The level of *SOD2* mRNA expression demonstrated a 3.5–4-fold increase upon transfection of the activator with sgRNA1, escalating to 15–20-fold with the use of all three sgRNAs (Figure 5A). Western blot analysis revealed a less significant rise of 1.9-fold and 8.7–8.8-fold in SOD2 protein amount, respectively (Figure 5B). The relative total superoxide dismutase activity of cellular homogenate increased by approximately 4.1 and 7.7 times, respectively (Figure 5D). Co-transfection of the cells with plasmids encoding four sgRNAs to the *CAT* gene promoter resulted in a 2–3-fold increase in its mRNA expression (Figure 5C).

The assessment of radioresistance (refer to Figure 5E) of HEK293T cells with *SOD2* and *CAT* overexpression, using the CFA method following irradiation at 2 Gy, shows an increase in cell survival, but only in the variant with the most extensive overexpression of *SOD2* and simultaneous overexpression of *CAT*. A similar pattern was observed after irradiation at a dose of 3 Gy, but a significant increase in survival was additionally observed in the cells overexpressing only *CAT*. After irradiation at 5 Gy, the cells exhibiting a 20-fold rise in *SOD2* expression alone also displayed enhanced radioresistance, similar to the results of experiments conducted with FMCA. Furthermore, the cells with moderate overexpression of *SOD2*, either individually or simultaneously with *CAT*, displayed a trend toward increased radioresistance. The results indicate that concurrent strong overexpression of *SOD2* and *CAT* results in enhanced cell survival after irradiation at all three doses, while strong overexpression of *SOD2* alone improves cell survival only after irradiation at the highest dose. These results suggest that an imbalance between successive stages of detoxification of ROS generated as a result of radiation-induced damages may be a contributing factor behind the varied outcomes of overexpression at different levels of stress factor exposure.

Balancing the antioxidant system during overexpression of superoxide dismutases by compensating for hydrogen peroxide overproduction has previously been performed in many studies. This was predominantly carried out in studies focusing on the effects of overexpression of cytoplasmic *SOD1*. For example, mouse embryonic fibroblasts (MEFs) with simultaneous overexpression of *SOD1* and *CAT* were more resistant to paraquat and hydrogen peroxide than cells with normal expression levels of both genes and overexpressing either *SOD1* or *CAT* alone [73]. *CAT* overexpression decreased ultraviolet-induced keratinocyte apoptosis to the same extent as simultaneous overexpression of *SOD1* + *CAT*. However, overexpression of *SOD1* alone did not affect this endpoint [74]. Compensation of H_2_O_2_ overproduction induced by *SOD2* overexpression was studied in the context of malignization processes. Simultaneous overexpression of *GPX1* enhanced the inhibitory effect of *SOD2* ectopic expression on the growth of pancreatic cancer cells in vitro and in vivo [49]. In experimental systems where *SOD2* overexpression resulted in an increase in the migration and invasion of tumor cells, this tumor-promoting effect was negated through the increased level of *CAT* mRNA [41,46]. In this study, it was demonstrated for the first time that simultaneous overexpression of *SOD2* and *CAT* enhances cellular radioresistance more effectively than overexpression of *SOD2* alone.

Analysis of the proliferation rate of the cells that formed colonies (Figure 5E) shows that there are no statistically significant differences between all transfection variants both without and after irradiation. At the same time, all exposure variants show a tendency to decrease the rate of cell proliferation when overexpressing *SOD2*. Thus, the effects of increased radioresistance of cell microcultures with *SOD2* overexpression detected using FMCA are a consequence of an increase in the proportion of surviving cells after irradiation rather than an increase in the proliferation rate of surviving cells.

The influence on proliferation not detected in this experimental system is not the only potential mechanism that could complicate the effect of SOD2 overexpression on cellular radioresistance. Analysis of the interactions of SOD2 with other proteins using the “InnateDB” platform [75] indicates its multifunctionality and almost uninvestigated potential involvement in other cellular systems. For example, SOD2 physically interacts with one of the component of the 40S subunit of ribosomes (CYP4F12) [76], which, in addition to its role in translation, is involved in the regulation of apoptosis [77] and with H2AX histone [78], which have major roles in the maintenance of genome stability [79]. Direct physical interaction with SOD2 was also found for one of the members of the cytochrome P450 family (CYP4F12) [80], and for aurora kinase A (AURKA) [81]. The interaction of SOD2 with TBL2, VAPA, STX7, NABP2, ALDH5A1, UBL4A and ZC3H11A was also shown as part of a large proteomic study [82]. Some of these proteins are known to be involved in cellular stress response. Transducin beta-like 2 protein (TBL2) plays a little-studied role in the endoplasmic reticulum stress response [83]. The aldehyde dehydrogenase ALDH5A1, in addition to its main function in cellular catabolism, also has a role in protection against oxidative stress [84]. Ubiquitin-like protein 4A (UBL4A) is a multifunctional signaling protein that is involved in the regulation of autophagy [85], and the single-stranded DNA binding protein NABP2 is one of the key proteins for the maintenance of genome stability [86].

All these experimentally demonstrated interactions do not indicate a direct role of SOD2 in the corresponding cellular processes and stress response systems but indicate that the observed changes in the radioresistance of cells with *SOD2* overexpression may not be the result of the basic enzymatic activity of the encoded protein alone.

## 3. Materials and Methods

### 3.1. Cell Lines

The HEK293T, HeLa and HT-29 cell lines were utilized for this experiment. They were cultured in DMEM/F12 medium (Paneco, Moscow, Russia) supplemented with 10% fetal bovine serum (FBS) (HyClone Laboratories, Logan, UT, USA) without antibiotics at 37 °C, 5% CO_2_. To prepare working suspensions, the cells were detached using a trypsin-EDTA solution (0.05%) with Hank’s salts (Paneco, Russia).

### 3.2. qRT-PCR and Primer Design

To determine the level of gene expression, the total RNA was extracted using the diaGene reagent kit (Dia-M, Moscow, Russia). RNA concentration was measured using the Qubit RNA HS Assay Kit on a Qubit fluorometer (Thermo Fisher Scientific, Waltham, MA, USA). The MMLV RT kit (Evrogene, Moscow, Russia) was used to perform the cDNA synthesis following the manufacturer’s instructions on a T100 PCR thermal cycler (Bio-Rad, Hercules, CA, USA). qRT-PCR was conducted using the “qPCRmix-HS SYBR” mastermix (Evrogen, Russia) and the CFX96 Touch Real-Time PCR Detection System (Bio-Rad, USA). The amplification reaction consisted of the following steps: 95 °C for 5 min, 40 cycles of 95 °C for 15 s, 58 °C for 15 s and 72 °C for 30 s. The expression level was calculated using the ΔΔCt method, relative to the geometric mean of the threshold cycles of *ACTB* and *GAPDH* genes [87]. The data were analyzed in CFX Manager (Bio-Rad, USA) and Excel (Microsoft Office, Redmond, WA, USA).

The primers intended for *CAT* mRNA and distinct transcriptional variants of *SOD2* were designed using the Primer-BLAST online tool [88] and are listed in Table 1. Figure 1 reveals the location of these primers on the *SOD2* transcription map. The primer sequences for *GAPDH* were borrowed from [89] (forward—ACACCCACTCCTCCTCCACCTTTG, reverse—GCTGTAGCCAAATTCGTTGTCATAC) and for *ACTB* from [90] (forward—GCGCGGCGGCTACAGCTTCA, reverse—CTTAATGTCACGCACGACGATTTCC). The primer specificity was verified by gel electrophoresis of amplification products in 1% agarose gel using a 100+ bp DNA Ladder (Evrogen, Russia).

### 3.3. Plasmids, sgRNA Design and Cloning

Transient gene overexpression was achieved by co-transfecting HEK293T cells with the plasmid “pR-CMV-dCas9-VPH-2A-TagGFP2” (Cellecta, Mountain View, CA, USA), which encodes dCas9 fused with a VPH activator [91] and plasmids encoding sgRNAs for the *SOD2* and *CAT* genes. The sgRNA design and cloning process followed the protocol outlined in a previous study [92] with the online tools “Cas-Designer’’ and “Cas-offinder” [93,94]. The “gRNA Cloning Vector BbsI ver. 2” plasmid served as the backbone and was obtained from Hodaka Fujii (Addgene plasmid #85586) [95]. The sgRNA sequences are summarized in Table 2. The genomic target location of sgRNAs is shown in Figure 1.

### 3.4. Transfection and Irradiation

Reverse transfection of the plasmids encoding the dCas9-VPH activator and sgRNAs into HEK293T cells was conducted in sterile 24-well culture plates using Lipofectamine^®^ 3000 Transfection Reagent (Invitrogen, Waltham, MA, USA). For this purpose, a transfection mixture was prepared according to the manufacturer’s protocol. For each well, 750 ng of plasmid encoding dCas9-VPH and 750 ng of sgRNA plasmids were used. The mixture of plasmids with transfection agents was spread on the growth surface of the wells. After 30 min, a suspension containing 200,000 cells in Opti MEM medium (Gibco, Billings, MT, USA) with 5% FBS was added accurately to each well. Forty-eight hours after transfection, the cells were seeded on fresh culture plates for irradiation and survival analysis. To irradiate the cells, a ^137^Cs *γ*-radiation source in the “Issledovatel” facility (Isotope, Zelenograd, USSR) was used at the dose rate of 0.74 Gy/min. Irradiation was performed 5 h after cell seeding, at doses of 2, 3 and 5 Gy.

### 3.5. Viability Assessment Using FMCA

Cell survival was assessed with the fluorometric microculture cytotoxicity assay (FMCA), following the protocol of [96]. For this purpose, the cells were seeded in 96-well plates (2000 cells per well) after transfection. The cells were then irradiated and subsequently incubated in standard conditions (37 °C and 5% CO_2_) for 72 h. Then, the medium was removed and 100 μL of fluorescein diacetate solution was added. After 40 min, incubation fluorescence intensity was measured at 485 nm (emission)/520 nm (excitation) on a microplate reader CLARIOstar Plus (BMG LABTECH, Ortenberg, Germany). Based on the fluorescence values, cell survival indices were calculated as a ratio of the average fluorescence in the treated cultures to the corresponding value in the untreated cultures.

### 3.6. Analysis of Clonogenic Survival and Proliferation Rate

The clonogenic survival was evaluated using the colony forming assay [97,98]. The proliferation rate of surviving cells was determined by estimating the area of the formed colonies. For this purpose, 48 h after transfection, the cells from each suspension (each transfection variant) were seeded into 24-well plates at the amount of 50 (for control), 100 (for 2 Gy), 200 (for 3 Gy) and 800 (for 5 Gy) cells per well, and 5 h after seeding, the cells were irradiated as described above. On the eighth day following irradiation, the cells were fixed with 96% ethyl alcohol, stained with Romanovsky azur–eosin dye (NPF ABRIS+, St. Petersburg, Russia), washed with distilled water and dried. The colony-forming units (CFUs) were quantified by visually counting the colonies. The results were expressed as the average number of colonies (in 12 wells) in each exposure/transfection variant relative to the non-irradiated control. CFU size, which reflects the cell proliferation rate, was measured using ImageJ software (National Institutes of Health, Bethesda, MD, USA) [99]. The area (mm^2^) of 24 randomly selected colonies from each variant was calculated to estimate the CFU size.

### 3.7. Western Blot Analysis

All stages of Western blot analysis were conducted in accordance with the previously outlined protocol [92]. The primary rabbit polyclonal antibodies PAB083Hu01 (Cloud-Clone Corp., Wuhan, China) were used to detect *SOD2*. The anti-Tubulin antibody ab4074 (Abcam, Waltham, MA, USA) was used as loading control. The chemiluminescent visualization was performed with recombinant anti-rabbit IgG VHH single-domain horseradish peroxidase conjugated secondary antibody ab191866 (Abcam, USA). The average relative protein content was calculated using four blots for each variant of transfection.

### 3.8. Analysis of SOD2 and Assessment of Total Superoxide Dismutase Activity

The total superoxide dismutase activity of cellular homogenate was assessed spectrophotometrically through the NBT photochemical assay [100]. The homogenate was prepared in pH 7.8 phosphate buffer using mechanical detaching of cells followed by 45 s sonication on ice using a Vibra Cell VCX 500 ultrasonic processor (Avantor, Radnor, PA, USA) at 40% power. Then, 3 wells of a 24-well plate were transfected and combined for each experimental variant and used for homogenate preparation 48 h after transfection. Each homogenate was assessed in 8 reactions using the microplate reader CLARIOstar Plus (BMG LABTECH, Germany). The relative total SOD activity was calculated using one of the standard formulas [101], with normalization to protein content, assessed with the Quick Start™ Bradford Protein Assay (BioRad, USA) [102].

### 3.9. Statistical Analysis

The ANOVA with post hoc Student–Newman–Keuls test was performed in the R programming language [103]. The ANOVA with Dunnett’s post hoc multiple comparisons was performed in Prism 8.0.1 software (GraphPad Software, San Diego, CA, USA).

## 4. Conclusions

In the current investigation using three human cell lines (HEK293T, HeLa, HT29), it was demonstrated that the expression of transcript variants 1 to 6 of the *SOD2* gene, encoding isoforms A to D, was 2–3 orders of magnitude greater than the expression of transcript variants 8 and 9, encoding isoform E. At the same time, despite the distance of the transcription start sites, both groups of transcripts are induced in response to oxidative stress triggered by exposure to ɣ-radiation at a dose of 2 Gy. The radiation induction of transcript variant 8 is the most pronounced. The expression of transcript variant 7 was not detected in the cell lines studied.

CRISPRa-mediated overexpression of key transcript variants of *SOD2* leads to an increase in the radioresistance of HEK293T cells, as determined using FMCA. Moreover, moderate overexpression (3.7–4.6-fold) had the greatest effect on culture viability after irradiation at 2 Gy, and strong overexpression (up to 22.5-fold) had an effect on culture viability after irradiation at 5 Gy. This result demonstrates the significance of the degree of overexpression, on the one hand, and the strength of the stressor, on the other hand, for the observed effect in terms of resistance. In a separate experiment, it was shown that strong overexpression of *SOD2* increased the clonogenic survival of the cells after irradiation at 5 Gy without increasing the proliferation of intact cells as well as surviving cells after irradiation.

Simultaneous overexpression of *SOD2* and *CAT* was found to enhance cell survival after irradiation more effectively than when these two genes were overexpressed separately. Together with the results of previous studies showing that ectopic expression of genes encoding enzymes that utilize hydrogen peroxide can suppress the procarcinogenic effects of superoxide dismutase overexpression, our findings support the potential efficacy of co-expression of these genes in reducing oxidative stress associated with various pathological processes.

## Figures and Tables

**Figure 1 ijms-24-17315-f001:**
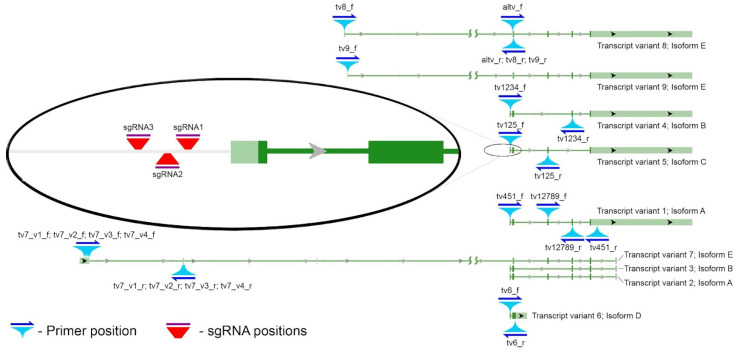
Primers’ and sgRNAs’ localization on the transcription map of the *SOD2* gene borrowed from the NCBI Gene database.

**Figure 2 ijms-24-17315-f002:**
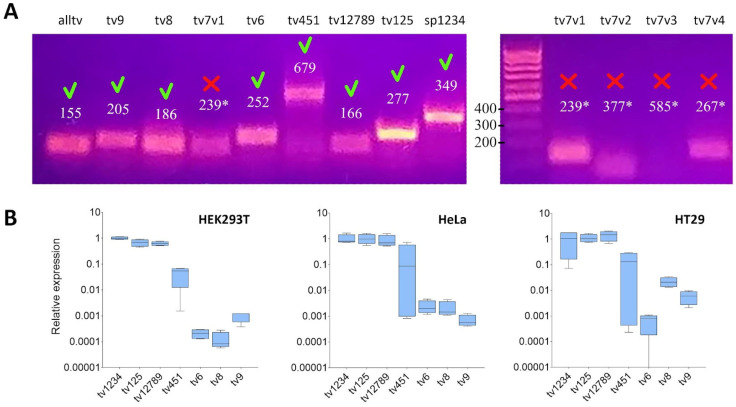
(**A**) Electrophoresis of amplification products of different transcription variants of the *SOD2* gene in HEK293T cells using the primers listed in Table 1. Green tick—expected amplification product. Red cross—the absence of the expected product. (**B**) Relative expression of transcripts and transcript groups of *SOD2* in HEK293T, HeLa and HT-29 cell lines. The average data from four biological replicates are presented. “alltv” is an amplicon obtained using primers specific to all transcription variants of the gene. “tv1234”, “tv125”, etc., were obtained using primers specific to the corresponding transcriptional variants. The amplicons “tv7v1”–“tv7v4” were obtained using primers specific to transcriptional variant 7. * expected amplicon size that does not match the electrophoresis result.

**Figure 3 ijms-24-17315-f003:**
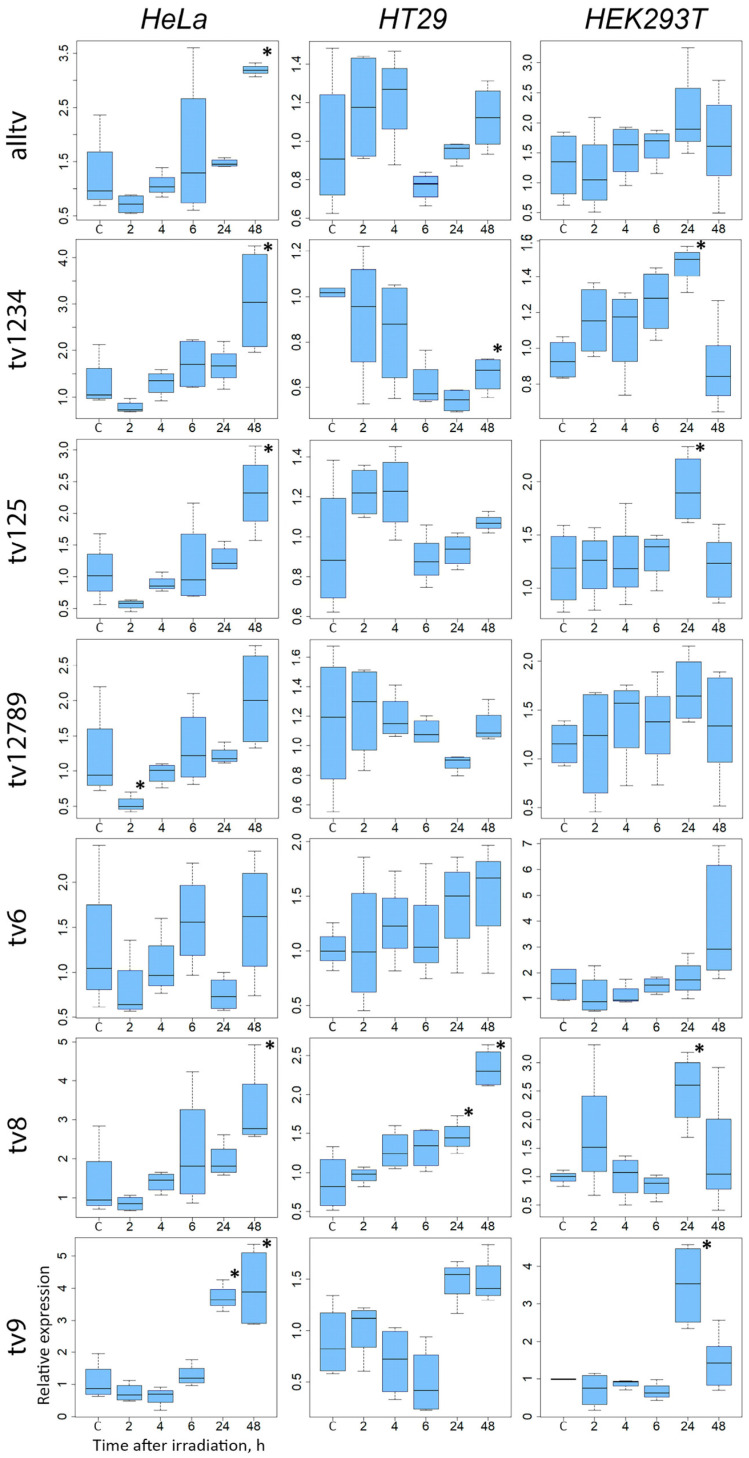
Relative expression levels of different transcript variants of *SOD2* after irradiation of HeLa, HT-29 and HEK293T cell lines with γ-irradiation at 2 Gy in four biological replicates of the experiment. C—untreated control. Significantly different variants (ANOVA, Student–Newman–Keuls test, *p* < 0.05) from non-irradiated control are indicated by asterisks. The designation of transcript variants is identical to that in Figure 2.

**Figure 4 ijms-24-17315-f004:**
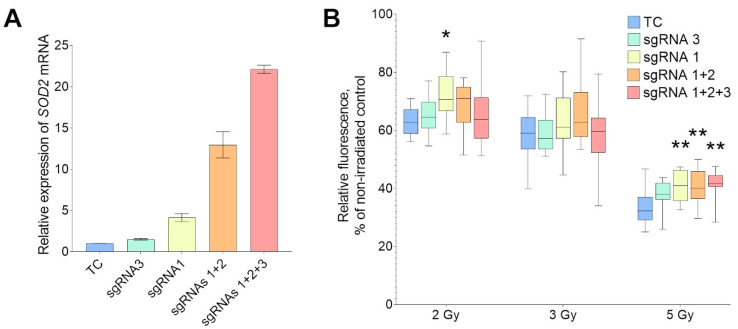
Radioresistance of HEK293T cells with different levels of ectopic *SOD2* expression evaluated using FMCA. (**A**) Relative expression of *SOD2* transcript variants 1–4 at 48 h after co-transfection of dCas9-VPH activator with plasmids encoding guide RNAs to the promoter of the proximal TSS of the gene (averages of 3 replicates). (**B**) The survival of transfected cells 72 h after irradiation at doses of 2, 3 and 5 Gy, analyzed using FMCA. Averaged data from 16 biological replicates per experiment variant are presented. “TC”—transfection control; “sgRNA 3”—the cells transfected with plasmids carrying dCas9-VPH activator and one sgRNA (distal) to the *SOD2* gene promoter; “sgRNA 1”—one proximal sgRNA; “sgRNA 1 + 2”—two sgRNAs; “sgRNA 1 + 2+3”—all three sgRNAs. * the differences with the corresponding TC variant are significant at *p* < 0.05; ** at *p* < 0.01 (ANOVA with Dunnett’s post hoc test).

**Figure 5 ijms-24-17315-f005:**
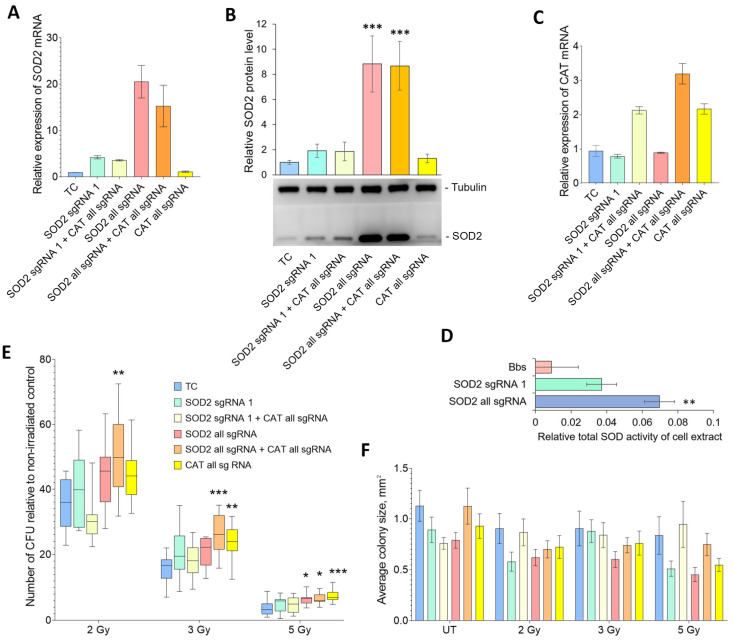
Radioresistance of HEK293T cells overexpressing *SOD2* and *CAT* evaluated through CFA. Relative expression of *SOD2* transcript variants 1–4 (**A**), results of Western blot analysis of total SOD2 protein (**B**) as mean relative protein expression and representative blots, *CAT* mRNA expression (**C**), and relative total SOD activity of cell homogenate (**D**) 48 h after co-transfection of dCas9-VPH activator with plasmids encoding guide RNA to the promoters of the corresponding genes. Averages of three repetitions of expression analysis, four replications of Western blot and eight replications of SOD activity analysis are presented. (**E**) Colony-forming ability of the cells with separate and simultaneous transient overexpression of *SOD2* and *CAT*. “TC”—transfection control. “SOD2 sgRNA1”—the cells transfected with plasmids carrying dCas9-VPH activator and one (proximal) sgRNA to the *SOD2* gene promoter; “SOD2 sgRNA 1 + CAT”—one sgRNA to the *SOD2* gene promoter and all sgRNAs to the *CAT* gene promoter; “SOD2 all sgRNA”—all sgRNAs to the *SOD2* gene promoter; “SOD2 all sgRNA + CAT all sgRNAs”—all sgRNAs to the *SOD2* gene promoter and all sgRNAs to the *CAT* gene promoter; “CAT all sgRNAs”—all sgRNAs to the *CAT* gene promoter. Averaged data from 12 biological replicates per experimental variant are presented. (**F**) Average area of colonies formed by surviving cells (24 colonies for each experiment variant). * the differences with the corresponding TC variant are reliable at *p* < 0.05; ** at *p* < 0.01, *** at *p* < 0.001 (ANOVA with Dunnett’s post hoc test).

**Table 1 ijms-24-17315-t001:** Experimentally validated primers specific for individual transcript variants of the *SOD2* and *CAT* genes.

Name	Forward Primer (5′-3′)	Reverse Primer (5′-3′)	Detectable Transcripts	Amplicon Length
altv	TCCGGTTTTGGGGTATCTGG	CGGTGACGTTCAGGTTGTTC	1–9	155
tv1234	ACTAGCAGCATGTTGAGCCG	TGAAACCAAGCCAACCCCAA	1, 2, 3, 4	349/466
tv125	CAGCACTAGCAGCATGTTGAGC	CAGTGCAGGCTGAAGAGCTATC	1, 2, 5	277
tv12789	CAGCCCTAACGGTGGTGGA	TTGTAAGTGTCCCCGTTCCTT	1, 2, 7, 8, 9	166
tv451	CACTAGCAGCATGTTGAGCC	TGACTAAGCAACATCAAGAAATGC	1, 4, 5	679
tv6	GCACTAGCAGCATGTTGAGC	CAGCCTGGAACCTACCCTTG	6	252
tv8	CACAGGAGAGTCGCCTTTCAG	GATCTGCGCGTTGATGTGAG	8	186
tv9	GCGGGCGTTTACTCTTAGCA	TCGGTGACGTTCAGGTTGTT	9	205
tv7_v1	AAAACTGTTGACGGACCTGGA	CTTTTCCCCTTCCCCTTGCTT	7	239 *
tv7_v2	AGGAGCATGTAACAAGTGGGG	GCCACCTCCGAAAAATTCCC	7	377 *
tv7_v3	ATGGGTCCTTTTGCTCTCGG	AAGTGGCCACCTCCGAAAAA	7	585 *
tv7_v4	CTTATGAGGGGCCACCGTTA	GGCCACCTCCGAAAAATTCC	7	267 *
CAT	TTCTGTTGAAGATGCGGCGA	TTCCTGTGGCAATGGCGTTA	-	83

* electrophoresis of the amplification product in an agarose gel indicates the absence of an amplicon of the expected size (see Figure 2A).

**Table 2 ijms-24-17315-t002:** SgRNA sequences employed in this study.

Gene	#sgRNA	sgRNA Sequence 5′-3′	Position Relative to TSS
*SOD2* (transcript variants 1–6/protein isoforms A–D)	1	CGCAGGGCACCCCCGGGGTT	129
	2	TGCCGTACACCCCGCGCCCA	172
	3	CCACTCAAGTACGGCAGAC	248
*CAT*	1	CAGAAGGCAGTCCTCCCGAG	80
	2	GCGCTAGGCAGGCCAAGAT	111
	3	TCCGGTCTTCAGGCCTCCTT	165
	4	GCGAGGCTCTCCAATTGCT	227

## Data Availability

Data is contained within the article.

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
