# Peer review of "Dose-Dependent Effect of Mitochondrial Superoxide Dismutase Gene Overexpression on Radioresistance of HEK293T Cells"

_ijms, 2023, doi:10.3390/ijms242417315_

Round 1

Reviewer 1 Report

Comments and Suggestions for Authors

In this study the authors have examined the effects of overexpression of genes encoding mitochondrial superoxide dismutase (SOD2) and catalase (CAT) on resistance of HEK293T cells to ionizing radiation. The results suggest that protective effect on cell survival is dependent on the degree of SOD2 overexpression and is enhanced by simultaneous overexpression of CAT. The manuscript is generally well written, the methodology is straightforward and the results are interesting. The limitations of this study are that it does not include the measurement of ROS production and determination of SOD and catalase activities.

There is one point that should be addressed and needs attention.

Figure 5: It is not clear whether quantification of SOD2 by Western blotting was made in several repetitions and whether it was statistically analyzed. Please, include the bar graph on relative expression of SOD protein as it is shown for relative expression of SOD2 mRNA.

The manuscript is acceptable for the publication providing that this point is corrected.

Author Response

Comments 1:

In this study the authors have examined the effects of overexpression of genes encoding mitochondrial superoxide dismutase (SOD2) and catalase (CAT) on resistance of HEK293T cells to ionizing radiation. The results suggest that protective effect on cell survival is dependent on the degree of SOD2 overexpression and is enhanced by simultaneous overexpression of CAT. The manuscript is generally well written, the methodology is straightforward and the results are interesting. The limitations of this study are that it does not include the measurement of ROS production and determination of SOD and catalase activities.

Response 1:

Dear Reviewer, Thanks for the careful reading of the manuscript and the high appreciation of the work. In connection with your remark and the comment of reviewer 2, the total relative activity of superoxide dismutases was analyzed by the only method available to us at the moment (using NBT photochemical assay) in cells with moderate and high overexpression of SOD2 mRNA. The result was added to the R1 version of the manuscript (Figure 5D).

Comments 2:

There is one point that should be addressed and needs attention.

Figure 5: It is not clear whether quantification of SOD2 by Western blotting was made in several repetitions and whether it was statistically analyzed. Please, include the bar graph on relative expression of SOD protein as it is shown for relative expression of SOD2 mRNA.

The manuscript is acceptable for the publication providing that this point is corrected.

In the original version of the manuscript, protein overexpression values were calculated based on only two membranes (independent repetitions of analysis), so there was no statistical analysis. In addition, we found our own mistake on the old version of the figure, the values of “1.2” and “1.4” were reversed. In the improved version, we added two more independent blots and got a similar result. In the new version of the manuscript, we calculated the relative amount of protein based on 4 membranes, plotted it, estimated the significance of the differences, and replaced the representative blot in the figure.

Reviewer 2 Report

Comments and Suggestions for Authors

The study by Tavleeva et al., assesses the radioresistant phenotype of 3 different cell lines (HEK293T, HeLa and HT29) in the presence and absence of over expression of various SOD2 transcript variants.  Although the premise of the paper was to have some insight for the contradictory literature on SOD2 over expression and different pathophysiological conditions, the way the data have been presented and results / conclusions were written did not deliver this intent.  Currently the manuscript is written basically descriptive listing of the findings with very little interpretation / or connection to the literature.  

The study needs some data causally linking the SOD2 activity and expression to the end points measured (or completely refuting the link by simply showing even if the activity levels did not significantly differ between transcript overexpressors, the protein levels are still causing effects on the end points measure - such as clonogenic efficiency)

Without such link, there is not really a trend considering the use of different IR doses as well as fluctuating numbers when SOD2 over expression was combined with CAT overexpression (and again there has been no activity data presented).

Comments on the Quality of English Language

A native speaker should go over the text to have smoother transitions perhaps (this is a very minor issue)

Author Response

Comments 1:

The study by Tavleeva et al., assesses the radioresistant phenotype of 3 different cell lines (HEK293T, HeLa and HT29) in the presence and absence of over expression of various SOD2 transcript variants.  Although the premise of the paper was to have some insight for the contradictory literature on SOD2 over expression and different pathophysiological conditions, the way the data have been presented and results / conclusions were written did not deliver this intent.

Response 1:

Thank you very much for your constructive criticism of the manuscript. We have tried to improve the work both experimentally and at the text level in accordance with your comments. One of the main ideas of the work was in demonstrations of the significance of the degree of overexpression, on the one hand, and the strength of the stressor, on the other hand, on the observed effect in terms of resistance. This was shown initially in two independent experiments (carried out at different times and using different methods) (Fig 4B и Fig 5E of R1 version of manuscript). So, in new version we added this point more clearly in conclusion section.

Comments 2:

Currently the manuscript is written basically descriptive listing of the findings with very little interpretation / or connection to the literature.

Response 2

The revision of the “discussion” was made in connection with your next comment (see below).

Comments 3:

The study needs some data causally linking the SOD2 activity and expression to the end points measured (or completely refuting the link by simply showing even if the activity levels did not significantly differ between transcript overexpressors, the protein levels are still causing effects on the end points measure - such as clonogenic efficiency)

Response 3:

Your remark raises a very important question, the experimental answer to which requires a lot of additional work (beyond the focus of this study): is it only the main activity of SOD2 affects the changes in survival that we and other researchers are observing, or is it due to other functions of this protein? As part of the improved version of the manuscript, on the one hand, we added the simplest measurement of the total activity of superoxide dismutases in the homogenate of cells with SOD2 overexpression to different degrees (new diagram 5D), and on the other hand, using the InnateDB platform, we found all the protein-protein interactions experimentally shown for SOD2. Based on this information, we expanded the discussion (Lines 383-405 of new manuscript version) with the main idea that SOD2 overexpression can potentially affect the radioresistance of cells not only by dismutation of the superoxide radical, but also by participating in other mechanisms of the cellular stress response. However, at the moment, these additional SOD2 features are still poorly understood.

Comments 4:

Without such link, there is not really a trend considering the use of different IR doses as well as fluctuating numbers when SOD2 over expression was combined with CAT overexpression (and again there has been no activity data presented).

Response 4:

Yes, in this work we do not see the full picture of cause-and-effect relationships. We've made this clearer by expanding the discussion, as I said above. But we see that at different levels of SOD2 mRNA, cells behave differently in response to different levels of stress. We can see that the amount of the SOD2 protein and now the total activity of all SODs changes with different mRNA expression. We also see that a change in the amount of catalase mRNA affects the radioresistance of cells with overexpression of SOD2 mRNA.

Comments 5:

A native speaker should go over the text to have smoother transitions perhaps (this is a very minor issue).

Response 5:

We showed the improved version of the manuscript to the staff translator (alas, he is not a native speaker, but we believe he was slightly improved the academic English of the manuscript).

Round 2

Reviewer 2 Report

Comments and Suggestions for Authors

The authors were fairly responsive to the critique.  Additional data and text in their discussion somewhat addressed the main issues (causal relation between activity and protein expression).  

I have one minor comment (which I clearly missed in my previous review) on statistical analysis.  I am assuming their analysis included an ANOVA before their post-hoc Dunnet test.  Because the Dunnet test is only used as a follow-up test to ANOVA. It cannot be used to analyze a stack of P values.  Please clarify

Author Response

The authors were fairly responsive to the critique.  Additional data and text in their discussion somewhat addressed the main issues (causal relation between activity and protein expression).  

I have one minor comment (which I clearly missed in my previous review) on statistical analysis.  I am assuming their analysis included an ANOVA before their post-hoc Dunnet test.  Because the Dunnet test is only used as a follow-up test to ANOVA. It cannot be used to analyze a stack of P values.  Please clarify.

Dear Reviewer,

Thank you again for your help in improving the manuscript. Yes, indeed, we forgot to indicate that ANOVA with Dunnett's post hoc test was used. We corrected this in the captions to Figures 4 and 5, as well as in section 3.9.